# OpenReview forum: "WildGuard: Open One-stop Moderation Tools for Safety Risks, Jailbreaks, and Refusals of LLMs"
_NeurIPS.cc/2024/Datasets_and_Benchmarks_Track — NeurIPS 2024 Track Datasets and Benchmarks Poster_

### Official Review · Reviewer_9fUj · 2024-07-21
**Great open-source initiative for safety moderation and evaluation**

**Rating:** 8
**Confidence:** 4
**Correctness:** high correctness.
**Clarity:** well-written

**Review:**

This is a top high-quality paper and I highly encourage accepting it.

**Strengths:**

- broad array of models evaluated
- novel model introduced
- novel data introduced (train, val and test)

**Additional Feedback:**

great work!

**Documentation:**

the documentation is great, specifically concerning human annotations.

**Limitations:**

are discussed but only in appendix. Should be moved to main text.

**Opportunities For Improvement:**

- W1 I think one problem is the disaggregated score. I think sometimes things get unsafe only if prompt and response are considered together not each alone. The evaluation would benefit from a general overall safety rating.
- W2 main paper misses limitations and ethical considerations IMO, only in the appendix.
- W3 the comparison of different moderation models is difficult. The authors discuss that there are multiple definitions for safety available, which I appreciate. On the other hand, other models have been trained on different taxonomies and hence have a different notion of safety. Hence, comparing them is not 1-1 fair. While there is nearly no way around it, discussing it would be highly beneficial to the reader.

**Relation To Prior Work:**

well established. Yet, more discussion on related work for taxonomy is needed, e.g. ALERT by Tedeschi et al. breaks taxonomy way more done, or MLCommons v0.5.

**Summary And Contributions:**

This paper introduces a novel text-based safety moderation tool including a new test set/benchmark. Their safety evaluation is three-fold, they distinguish between prompt safety, response safety, and response refusal.

---

> ### Author Rebuttal · Authors · 2024-08-15
>
> Thank you for acknowledging our contributions in introducing the model, data, and broad experimental results for safety moderation. We appreciate your thoughtful suggestions for improving our paper. Here are our responses to your comments:
>
> ---
>
> > I think sometimes things get unsafe only if prompt and response are considered together not each alone. The evaluation would benefit from a general overall safety rating.
>
> We fully agree that there can be defeasible cases where harm arises only when both the prompt and the response are considered together. We intend to address this challenge in future work, and we thank you for highlighting this important consideration.
>
> > Main paper misses limitations and ethical considerations, only in the appendix.
>
> Due to the page limitations, we included these contents in the appendix. We will include them into the main paper within the additional pages available for the camera-ready version.
>
> > Different models have been trained on different taxonomies and hence have a different notion of safety. Hence, comparing them is not 1-1 fair. While there is nearly no way around it, discussing it would be highly beneficial to the reader.
>
> Thank you for this suggestion. Indeed, we also recognize the concept of safety can vary across models, making direct comparisons challenging. To avoid overemphasizing certain models, we conducted a comprehensive evaluation by including diverse evaluation benchmarks in our experiments. We agree that further discussion on this topic in the limitations section would be beneficial, and we will add it to the camera-ready version.
>
> > Yet, more discussion on related work for taxonomy is needed, e.g. ALERT by Tedeschi et al. breaks taxonomy way more done, or MLCommons v0.5.
>
> Thank you for these references. We are happy to expand the risk taxonomy so that our model and benchmark can cover more risk categories. We will discuss more about the taxonomy in the limitations section in the camera-ready version.

---

> > ### Comment · Reviewer_9fUj · 2024-08-19
> >
> > thanks for your response and clarifications to my points raised and for integrating them into the camera-ready version.

---

### Official Review · Reviewer_Tk4P · 2024-07-23
**Overall a strong contribution that includes a valuable dataset and safety model**

**Rating:** 7
**Confidence:** 2

**Review:**

On the one hand, the experimental results demonstrate the robustness and effectiveness of WILDGUARD across various benchmarks. On the other hand, the authors have carefully built a large dataset, WILDGUARDMIX, that enables training/evaluating not only on prompt harmfulness and response harmfulness, but also on response refusal. Overall, this paper makes a valuable contribution for the community and lays the groundwork for future research in LLM safety moderation.

**Strengths:**

1. WILDGUARD outperforms existing open-source content moderation tools like Llama-Guard2, especially in discerning harm in adversarial prompts (i.e., jailbreaks) and evaluating model refusals.
2. The authors made a lot of effort curating a high-quality safety dataset that comprises 92K labelled examples across various safety risk categories. It will be a valuable resource for training and evaluating future moderation tools.
3. The authors provide an extensive evaluation of WILDGUARD against various baselines on multiple open-source benchmarks. They show a strong performance across several key metrics.
4. WILDGUARD and its dataset are publicly available.

**Additional Feedback:**

No additional comments.

**Clarity:**

The paper is well-written and clearly structured. However, presenting the evaluation results (Section 2) before describing the datasets (Section 3.2) slightly disrupts the flow and makes it harder to follow if read in a one pass through.

**Correctness:**

There are no concerns about the correctness of the methodology or results. The experiments are well-designed, and the results are convincingly presented.

**Documentation:**

The work includes appropriate documentation.

**Ethics:**

I do not see any additional ethical concerns arising from this work. The paper addresses safety and moderation.

**Limitations:**

Limitations are well addressed.

**Opportunities For Improvement:**

1. Large parts of the dataset are generated by GPT-4, which might pose questions regarding the dataset's diversity and potential biases introduced by the model.

**Relation To Prior Work:**

The paper provides a comprehensive comparison with existing safety moderation tools and benchmarks.

**Summary And Contributions:**

The authors introduce WILDGUARD, a moderation tool for LLM safety that helps identify malicious intents in user prompts, detect safety risks in model responses, and determine the model's refusal rate. WILDGUARD offers a range of 13 safety risk categories and improved robustness for identifying adversarial jailbreaks. WILDGUARD also comes with its safety moderation dataset WILDGUARDMIX. The dataset contains a total of 92k labeled examples that are split into train and test sets. The test set comprises 5k samples that are human-annotated. WILDGUARD is evaluated on multiple safety moderation benchmarks and outperforms existing open-source moderation models. WILDGUARD matches and sometimes exceeds GPT-4 performance.

---

> ### Author Rebuttal · Authors · 2024-08-15
>
> Thank you for recognizing our contributions in introducing data, developing a model, and providing extensive experimental results.
>
> To mitigate bias and improve the diversity of our dataset, we employed several techniques in our data construction pipeline (as described in Section 3.1.1, Appendix C.1), including: 1) incorporating diverse topics relevant to safety, 2) randomly sampling in-context examples to enhance the diversity of the synthesized examples, 3) running a minhash lexical deduplication based on 3-gram overlap, and 4) employing the WildTeaming framework to discover diverse jailbreak tactics when generating adversarial prompts.
>
> We agree that, despite our efforts to mitigate the bias, LLM-generated data might still contain biases that we couldn’t fully remove, and we appreciate you for bringing this to our attention. We hope to further address this issue in the future works.
>
> We also thank you for your comment about the presentation order of the sections (Section 2 and 3). Our purpose was to motivate readers by addressing the problem that existing open moderation tools struggle to detect adversarial cases effectively. We understand that readers who might not be familiar with this topic could find it difficult to read in one pass, and we will revisit the flow to make it more accessible.

---

> > ### Comment · Reviewer_Tk4P · 2024-08-30
> >
> > I appreciate the authors' efforts to address my concerns, and since my rating is positive, I will keep it unchanged.

---

### Official Review · Reviewer_rRrT · 2024-07-27
**Valuable moderator dataset**

**Rating:** 7
**Confidence:** 4
**Correctness:** Yes
**Clarity:** Yes

**Review:**

**Significance:** Overall, I think the contribution is decent enough, especially the 92K data samples in WildGuardMix carefully crafted to train and evaluate in multiple dimensions of safety-13 risk categories as well as the spread of prompts (harmful, benign, adversarial, in-the-wild). WildGuard is a Mistral 7b model trained on WildGuardTrain.

- Clarity: The paper is easy to read and clearly written.

- Originality: In my understanding, the work is original as I have not seen such a big and quality dataset till date.

**Strengths:**

The following are the strengths:

- WildGuard introduces a significant advancement in moderation tools by tackling jailbreak prompts and building resistance to exaggerated safety/incorrect refusals. The demonstrated improvement is as high as 25.3 % in detecting refusals and reducing the attack success rate from around 80% to 2%.

- The proposed training set WildGuardTrain is (probably) the largest moderator training dataset with wide coverage over 13 risk categories.

**Additional Feedback:**

NA

**Documentation:**

Yes

**Opportunities For Improvement:**

I do not have any major concerns, just a few suggestions:

Since the dataset is a (most) valuable contribution of the work, I think the prompt construction process is not fully clear to me. It is still not very clear how the exact process of collecting _vanilla harmful synthetic prompts_ and _adversarial benign and harmful synthetic prompts_ was carried out. I believe citing the WildTeaming work is okay, but detailing the concrete process is still necessary, especially when this aspect is an integral part of the approach.

**Relation To Prior Work:**

Yes

**Summary And Contributions:**

The paper introduces WildGuard, a new guardrails/moderation tool. WildGuard fills the loopholes of existing moderation tools such as Llama-Guard2, which work well on straightforward queries but suffer from jailbreak attacks/crafted prompts. To construct WildGuard, the critical step was to construct a quality dataset referred to as WildGuardMix which is a multi-task moderation dataset with 92K labeled examples covering vanilla and adversarial jailbreak prompts. To test the moderation quality (evaluation phase), disjoint 5K samples are also constructed. The two subsets of WildGuardMix are referred to as WildGuardTrain and WildGuardTest. WildGuard trained on WildGuardTrain shows superior performance when compared with open-weight moderation tools and at part with GPT-4 performance.

---

> ### Author Rebuttal · Authors · 2024-08-15
>
> Thank you for your thorough review and for considering the significance of our data and model.
>
> Regarding your comment on the data construction details: we will include more specific descriptions, particularly concerning the creation of adversarial prompts. Thank you for your great suggestion.

---

### Decision · Program_Chairs · 2024-09-26

**Decision:**

Accept (Poster)

**Comment:**

- Summary:
    - (borrowed Reviewer Tk4P) The authors introduce WILDGUARD, a moderation tool for LLM safety that helps identify malicious intents in user prompts, detect safety risks in model responses, and determine the model's refusal rate. WILDGUARD offers a range of 13 safety risk categories and improved robustness for identifying adversarial jailbreaks. WILDGUARD also comes with its safety moderation dataset WILDGUARDMIX. The dataset contains a total of 92k labeled examples that are split into train and test sets. The test set comprises 5k samples that are human-annotated. WILDGUARD is evaluated on multiple safety moderation benchmarks and outperforms existing open-source moderation models. WILDGUARD matches and sometimes exceeds GPT-4 performance.
- Overall:
    - This paper has consistent positive reviews. Strengths are the high-quality and large size of the 92K dataset in multiple dimensions of harmful, benign, adversarial, and in-the-wild. The dataset and the trained moderation models are also publicly released. The paper is clearly well-written.
    - Potential risks of LLM-generated datasets should be discussed in the limitation section, as well as diverse taxonomies of safety across different models.